# Elastic Dynamic Sling on Subluxation of Hemiplegic Shoulder in Patients with Subacute Stroke: A Multicenter Randomized Controlled Trial

**DOI:** 10.3390/ijerph19169975

**Published:** 2022-08-12

**Authors:** Min Gyun Kim, Seung Ah Lee, Eo Jin Park, Min Kyu Choi, Ji Min Kim, Min Kyun Sohn, Sung Ju Jee, Yeong Wook Kim, Jung Eun Son, Seo Jun Lee, Keum Sun Hwang, Seung Don Yoo

**Affiliations:** 1Department of Rehabilitation Medicine, Kyung Hee University Hospital at Gangdong, Seoul 05278, Korea; 2Department of Physical Medicine and Rehabilitation, Graduate School, Kyung Hee University, Seoul 02447, Korea; 3Department of Rehabilitation Medicine, Chungnam National University Hospital, Daejeon 35015, Korea; 4Department of Rehabilitation Medicine, Chungnam National University Sejong Hospital, Sejong 30099, Korea; 5Department of Medicine, AgeTech-Service Convergence Major, Kyung Hee University, Seoul 02447, Korea

**Keywords:** stroke, infarction, shoulder subluxation, orthoses, hemiplegia, rehabilitation, shoulder pain

## Abstract

Background: Shoulder subluxation occurs in 17–64% of hemiplegic patients after stroke and develops mostly during the first three weeks of hemiplegia. A range of shoulder orthoses has been used in rehabilitation to prevent subluxation. However, there is little evidence of their efficacy. AIM: This study aimed to investigate whether there is a difference in the subluxation distance, pain, and functional level of the hemiplegic upper extremity among patients with two different shoulder orthoses. Design: This is a prospective, randomized controlled trial with intention-to-treat analysis. SETTING: Multicenter, rehabilitation medicine department of two university hospitals in South Korea. Population: Forty-one patients with subacute stroke with shoulder subluxation with greater than 0.5 finger width within 4 weeks of stroke were recruited between January 2016 and October 2021. Methods: The experimental group used an elastic dynamic sling while sitting and standing to support the affected arm for eight weeks. The control group used a Bobath sling while sitting and standing. The primary outcome was to assess the distance of the shoulder subluxation on radiography. The secondary outcomes were upper-extremity function, muscle power, activities of daily living, pain and spasticity. Result: The horizontal distance showed significant improvement in the elastic dynamic sling group, but there were no significant differences in the vertical distance between the elastic dynamic and Bobath sling groups. Both groups showed improvements in upper-extremity movements and independence in daily living after 4 and 8 weeks of using shoulder orthoses, and the differences within the groups were significant (*p* < 0.05). However, there were no significant differences in upper-extremity movements and independence in daily living between the two groups. Conclusions: The subluxation distance showed better results in the elastic dynamic sling, which has both proximal and distal parts, than in the Bobath sling, which holds only the proximal part. Both shoulder orthoses showed improvements in the modified Barthel index, upper-extremity function, and manual muscle testing.

## 1. Introduction

In stroke patients, shoulder subluxation is a common complication. Weakness of the upper extremity of the affected side and the weight of the dependent arm cause a downward displacement of the humeral head from the shallow glenoid fossa, causing shoulder subluxation [1]. The etiopathogenesis is unclear, but it has been suggested that weak muscles around the shoulder joint interrupt the mechanical integrity and stability of the joint, resulting in a palpable gap between the acromion and humeral head. In the first three weeks of hemiplegia, the affected arm is flaccid or hypotonic; hence, the shoulder muscles cannot anchor the humeral head within the glenoid cavity. The incidence of shoulder subluxation on the hemiplegic side ranges from 17% to 64% [2,3,4,5].

Stroke can cause shoulder subluxation and may lead to hemiplegic shoulder pain, resulting in shoulder contracture and secondary irreversible damage to the muscles, ligaments, joint capsules, nerves, and blood vessels. Pain and joint contracture caused by shoulder subluxation can have a negative impact on the recovery of upper-extremity function in patients with stroke [6]. It can lead to serious limitations in activities of daily living, balance, mobility, and upper-limb and hand functions. It is associated with a higher incidence of depression, both during and after rehabilitation [7,8].

The underlying hypothesis for the association between shoulder subluxation and pain is that gradual stretching of the capsule and tendons causes them to become ischemic and painful. In addition, the weight of the arm and sustained stretching of the soft tissues can cause damage and inflammation [9].

To prevent and treat shoulder subluxation, arm rest, shoulder orthosis, shoulder taping, and functional electrical stimulation, botulinum toxin, peripheral nerve stimulation (PNS), transcutaneous electrical nerve stimulation (TENS), and neuromuscular electrical stimulation (NMES) are being performed [10,11,12,13,14,15]. Among them, orthoses may be implemented to provide a low-load prolonged stretch to prevent length-associated changes in muscles and connective tissue that can limit the function of the affected limb after stroke [16]. An orthosis is a removable device that immobilizes joints for therapeutic purposes by applying a prolonged static stretch to the muscles. The proposed benefits of orthoses in individuals with neurological impairments include decreasing spasticity, improving function, preventing contracture, minimizing pain, and decreasing swelling [17].

Various types of shoulder orthoses are used to prevent and treat subluxations. Based on a 2005 Cochrane review [10], there was insufficient evidence to conclude whether shoulder slings could prevent vertical subluxation or decrease shoulder pain. The authors recommended that randomized controlled trials be conducted to evaluate the efficacy of devices to support the shoulder. This expert consensus also recommended that such devices should be trialed immediately when the patient could be positioned upright and continued for a period of four to six weeks as research is lacking on the immediate post-stroke period.

In this study, two different shoulder orthoses were used in patients with subacute stroke. The elastic dynamic shoulder sling, a new orthosis with proximal and distal attachments, was compared with the commonly used Bobath roll sling. The purpose of this study was to investigate whether there is a difference in the subluxation distance, pain, and upper-extremity function between the two shoulder orthoses.

## 2. Materials and Methods

### 2.1. Design

This was a prospective, randomized, controlled, multicenter trial. The patients who experienced stroke for the first time and receiving inpatient treatment with dislocations greater than 0.5 finger width were recruited between January 2016 and October 2021 at the Rehabilitation Department of the Kyunghee University Hospital at Gangdong and Chungnam National University Hospital.

### 2.2. Randomization

The scientific validity of the clinical trial was ensured by maximizing the comparability of the experimental (elastic dynamic shoulder sling group) and control (Bobath sling group) groups by implementing a randomization method and preventing the interference due to the subjectivity of the research team. Using a random function in Excel, a stratified randomization code was generated with sex and institution as stratification variables. The ratio of the test and control groups was 1:1.

### 2.3. Participants

Patients were included if they were within 4 weeks of their first stroke, had a shoulder dislocation greater than 0.5 finger width and has cognitive function with the ability to express pain. Patients were excluded if they had shoulder weakness before stroke (which may be due to spinal cord injury and myopathy), inability to evaluate pain (as is seen in patients with total aphasia and cognitive decline), history of shoulder joint disease before stroke, and age < 18 years. The average age of the participants was 64.19 ± 13.48 years. The study population consisted of 26, 15, 2 and 39 patients with infarction, hemorrhage, brain stem lesions, and non-brain stem lesions, respectively.

### 2.4. Intervention

The experimental group received elastic dynamic shoulder sling (Figure 1) and the control group received Bobath sling (Figure 2) to support affected upper extremity.

Both groups wore their orthoses immediately after transfer to the Department of Rehabilitation Medicine within four weeks from stroke onset. They wore the orthoses for a period of 8 weeks during the active time of the day, but not when lying in bed or during formal therapy sessions. All patients, independent of the assigned study group, under-went the same standard rehabilitation program. The therapy program focused on avoiding complications related to the severely impaired upper limbs.

Examinations and evaluations, including radiography, were performed during clinical follow-up visits. The timing of the procedures are as follows (Figure 3).

T1 (immediately transferred to the Department of Rehabilitation Medicine within four weeks of stroke onset).

Radiography before and after wearing the shoulder brace, Fugl-Meyer assessment (FMA) scale, manual muscle testing (MMT), pain, modified Ashworth scale (MAS), and Korean-modified Barthel index (K-MBI) were assessed.

T2 (four weeks after T1, at the time of discharge from the Department of Rehabilitation Medicine).

Radiography before wearing the shoulder brace, FMA scale, MMT, pain, MAS, and K-MBI were assessed.

T3 (at outpatient follow-up, four weeks after T2)

Radiography before wearing the shoulder brace, FMA scale, MMT, pain, and K-MBI were assessed.

Types of sling

(1)Elastic dynamic shoulder sling: The shoulder device consists of three main parts: (1) a shoulder belt placed over the affected shoulder, (2) chest belt, and (3) wrist belt on the affected side. It uses elastic material to lift the deltoid and fix the scapula so that it can be adducted and retracted (Figure 1).(2)Bobath sling: The shoulder device consists of three main parts: (1) a foam roll placed in the axillary region of the affected shoulder, (2) a figure-8 pattern that connects the shoulder blades, and (3) a horizontal strap made of similar material that encircles around the chest (Figure 2).

### 2.5. Outcomes

(1)Primary outcomes

Subluxation distance: Measured with true anteroposterior X-ray. It brings the scapula of the injured side parallel to the X-ray plate. This avoids overlapping of the humerus head and the glenoid.

After each participant was seated on a chair, a true anteroposterior (AP) simple radiographic examination of both shoulder joints was performed in an upright posture, with the arm in a neutral position hanging down under gravity.

(2)Secondary outcomes

FMA: To evaluate the recovery of motor function in stroke patients, the upper-extremity motor function was evaluated using the FMA scale. The maximum score is 100 points, with 66 points for upper-extremity motor function and 34 for lower-extremity motor function. In this study, only upper-extremity motor function was assessed of three points, with 0 being unable to perform, 1 being partially able to perform, and 2 being completely capable of performing. This test is known to have high reliability between test and retest and high inter-examiner reliability and validity [18].

K-MBI: The degree of dependence of the patient when performing activities of daily living was evaluated in five categories: complete independence, little help, moderate help, much help, and complete dependence. The evaluation consisted of ten areas: eating, dressing, dressing up, bathing, moving in a chair/bed, moving and using the toilet, walking (or moving a chair car), using stairs, and controlling bowel movements [19].

Pain: The degree of shoulder pain at each time point was indicated by a visual analog scale (VAS, 0–10), which is commonly interpreted as a reasonably valid report of subjective pain. Each participant was asked to rate the presence and degree of pain in the affect-ed shoulder on a scale of 0 (no pain experienced) to 10 (worst pain imaginable) during evaluation.

MAS: It is the most universally accepted clinical tool that is used to measure increase in muscle tone. Spasticity was defined by Jim Lance in 1980 as a velocity-dependent increase in muscle-stretch reflexes associated with increased muscle tone as a component of the upper motor neuron syndrome [20].

MMT: It is the most commonly used method for documenting impairments in muscle strength [21]. The muscle power of the shoulder deltoid muscles was examined. Shoulder forward flexion and abduction were tested by manual muscle testing procedure. Average was recorded.

### 2.6. Data Analysis

Three analysts measured and analyzed the radiographs in a random order to reduce measurement bias. Distance measurements of shoulder subluxation from a single radiograph were used, as described by Brooke et al. [22]. The central point of the glenoid fossa of the scapula was determined by marking the most distant vertical and horizontal edges. Height and width measurements were then bisected to determine the location of the central point of the glenoid fossa. The central point of the humeral head was determined by measuring the greatest distance that could be horizontally obtained across the head. This line was bisected to provide the central point of the humeral head. The inferior acromial point was determined by identifying the most inferior point on the acromial and lateral surfaces of the acromioclavicular joint. The vertical distance (VD) was measured from the acromial point to the central point of the humeral head. The horizontal distance (HD) was measured from the central points of the humeral head and the glenoid fossa (Figure 4).

### 2.7. Statistical Analysis

Statistical analysis was performed using Statistical Package for Social Sciences (version 25.0; SPSS Inc., Chicago, IL, USA). The analysis was conducted by an independent scientist and statistician. *p*-value of <0.05 was considered significant.

Independent-samples *t*-test was used to confirm differences in the degree of subluxation between the groups such as the difference between the radiographic test results in T2 − T1 (ΔT1) and T3 − T2 (ΔT2) and the difference in the radiographic test results before and after wearing a shoulder brace in T1. The difference in radiological examination results in T2 − T1 (ΔT1) and T3 − T2 (ΔT2) was calculated. A linear mixed model was used to confirm changes within the group over time [23].

With the power set at 80% and an overall *p* < 0.05, we needed 21 subjects per group. To allow for dropouts, we planned to recruit 36 participants per group. Post hoc power analysis showed that group sample sizes of 21 and 20 achieved 80.940% power to reject the null hypothesis of equal means when the population mean difference was μ1 − μ2 = 2.28 − (−0.08) = 3.08, with standard deviations of 3.11 for group 1 and 3.66 for group 2, and with a significance level (alpha) of 0.050 using a two-sided two-sample unequal variance *t*-test. Effect size was 0.909 [24].

## 3. Results

The flow of participants during the trial is summarized in Figure 5. From January 2016 to October 2021, 241 patients with stroke were assessed for eligibility. A total of 125 patients did not meet the inclusion criteria, and 44 declined to participate. A total of 72 patients participated in this study, of whom 31 dropped out for reasons such as refusal to wear the Bobath sling (1.3%), stroke recurrence (1.3%), change in the Bobath sling to an elastic dynamic sling (2.7%), and early discharge and follow-up loss due to Coronavirus disease (COVID-19) (37.5%). Finally, 41 patients were included in the final study.

Comparisons and statistical analyses between the groups were performed at baseline, four weeks, and eight weeks. Table 1 shows the baseline characteristics of the participants.

The average age of the participants was 64.19 ± 13.48 years. The study population consisted of 26 patients with infarction, 15 with hemorrhage, two with brain stem lesions, and 39 with non-brain stem lesions. There were no significant differences (*p* < 0.05) between the two groups in terms of baseline characteristics, including sex, age, stroke, location of the lesion, and baseline measurements.

Comparisons of the primary outcomes are shown in Table 2 and Table 3.

There were no significant differences in the vertical distance between the elastic dynamic sling and Bobath sling groups. Horizontal distance was significantly reduced in the elastic dynamic sling group compared to that in the Bobath sling group at eight weeks after sling usage (*p* = 0.006). As shown in Table 4, the horizontal distance of the affected shoulder gradually increased in the Bobath sling group.

Comparisons of secondary outcomes within the groups are shown in Table 5.

All participants demonstrated an increase in MBI, FMA scale, and MMT of the shoulder after four weeks and eight weeks of intervention without significant improvement in pain.

The comparisons between the groups are shown in Table 6.

There were no significant differences in MBI, FMA scale, MMT of the shoulder, and pain.

## 4. Discussion

The results showed a significant difference in the horizontal subluxation distance at 8 weeks compared to 4 weeks, which indicates that the effectiveness of the elastic dynamic sling increased with longer periods of use.

The Bobath sling used in the control group only provided proximal support. In light of the study results, the main benefit of the Bobath roll is the alignment of the upper limb as a whole, avoiding flexion and internal rotation [22,25]. The arm is supported in a pattern of abduction and extension; therefore, flexor spasticity throughout the whole upper limb is potentially reduced. The limb is free of function and is important for balance. This position allows for increased motor activity, symmetry, and bilateral upper-extremity activity. The support remains aesthetically acceptable and covered by garments [26].

Radiological evidence indicated that the Bobath sling caused significant distraction of the humerus in the horizontal plane. Other studies on the Bobath shoulder sling also identified, through the use of radiographs, that the Bobath sling produced a significant lateral displacement of the head of the humerus [27,28]. This study showed similar results as those of a previous study, which showed that horizontal distance gradually increased over time (Table 4).

The elastic dynamic shoulder sling showed a similar effect on vertical subluxation as that of the Bobath sling. It is made of a stretchable material, due to which it can adjust shoulder subluxation in both the horizontal and vertical axes. Therefore, it was possible to correct the deflection in the horizontal direction, which the Bobath sling could not. In addition, the proximal and additional distal support allows the patient to freely use their hands and wrists while wearing an orthosis during rehabilitation. This result correlates with that of a systematic review by M. Nadler in 2017 [14] on shoulder orthosis, which showed that orthoses with proximal and distal attachments are more effective in preventing shoulder subluxation.

In a previous study, horizontal shoulder subluxation was found to cause supraspinatus tendinitis. The supraspinatus tendon is one of the major sites of soft tissue injuries and lesions. It may cause more pain and poor upper-limb motor function, combined with impaired sensation and shoulder spasticity [29]. In this study, the horizontal distance gradually increased in the Bobath sling group. However, no significant improvement in pain was observed in either group. The VAS score, used in this study as an indication of pain, was a subjective index. It was difficult to make a clear comparison of pain before and after wearing the shoulder orthosis. This is because most patients with stroke have cognitive impairment. The VAS score that was expressed before wearing the orthosis was not provided to the patients during the survey after wearing it.

Both groups showed improvements in upper extremity-function and activities of daily living (Table 5). The goal of rehabilitation therapy for patients with hemiplegia is to restore independence in limb movements and everyday activities. We selected the upper-extremity Fugl-Meyer assessment (FMA) to reflect improvements in upper-limb activity, and the Korean-modified Barthel index (K-MBI) to measure independence in performing everyday activities.

However, there is an uncertainty regarding how support devices can improve mobility. Several factors may have influenced this finding. One reason for this is that the device could maintain the paralyzed upper limb in a reflex inhibition pattern, which could prevent the development of inefficient movement and ensure that the normal position is maintained in the paralyzed limb. Normal position of the paralyzed upper limb may contribute to functional recovery. The application of dynamic shoulder sling and Bobath sling may encourage patients to exercise properly.

Rehabilitation in this study combined physical exercise with position correction. Therefore, we cannot conclude that the elastic dynamic sling and the Bobath sling could improve limb and body activity function by themselves, but they could be beneficial when combined with physical exercise in the recovery and rehabilitation progress.

In addition, there are no standard measurements for evaluating shoulder subluxation. New methods are being developed to accurately measure subluxation, such as diagnostic ultrasound, using clearly defined landmarks. Although research on this is ongoing, the measurement method using ultrasound is not yet the standard measurement method.

By preventing the common complications of subluxation and hemiplegic shoulder pain, patients may be able to participate more extensively in upper-limb rehabilitation, enabling them to maximize their functional recovery and independence.

The limitations of this study should be noted for correct interpretation of the present results. First, 27 patients were lost to follow-up. The coronavirus disease (COVID-19) pandemic was the primary cause. Second, the patients who participated in this study were stroke patients, and cognitive deficits were present in over 70% of the stroke survivors [30]. When conducting a questionnaire study on pain at follow-up after eight weeks of wearing the brace, the patient did not inform the value of the initial response. Therefore, it was difficult to accurately judge whether there was an improvement or deterioration compared to the results based on the previous questionnaire. Third, there is no precise method for measuring shoulder subluxation. Lastly, there is a lack of studies on long-term follow-up of patients with hemiplegia and horizontal shoulder subluxation. Further studies are required to address this issue.

## 5. Conclusions

In a previous study, shoulder orthosis with both distal and proximal parts showed better effects on patient function and pain than orthosis with only the proximal part [10]. In this study, the distance of the horizontal subluxation was adjusted better in the elastic dynamic shoulder sling, which has both proximal and distal parts, than in the Bobath sling, which holds only the proximal part. It may reduce the incidence of supraspinatus tendinitis and may reduce the pain. Both shoulder orthoses showed improvements in MBI, upper-extremity function, and MMT. The application of shoulder orthoses could also improve upper-limb motor function and daily activities in stroke patients. However, no clear differences were observed between the two groups and further research is required.

## Figures and Tables

**Figure 1 ijerph-19-09975-f001:**
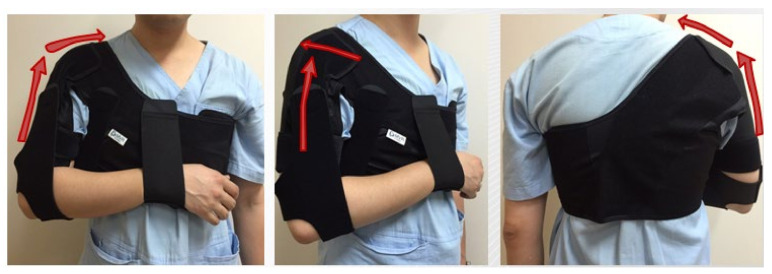
Elastic dynamic shoulder sling.

**Figure 2 ijerph-19-09975-f002:**
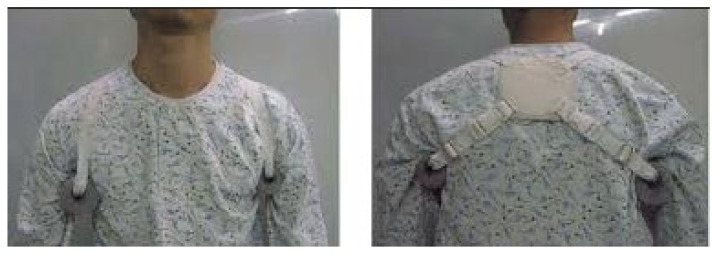
Bobath sling.

**Figure 3 ijerph-19-09975-f003:**
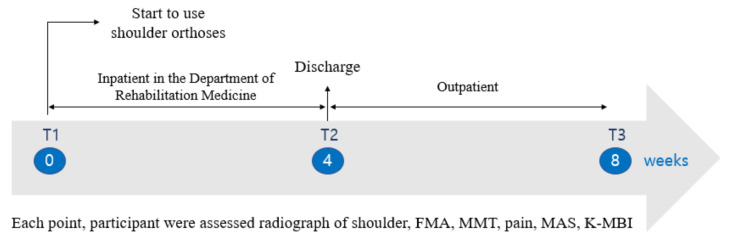
Timeline of assessment and follow up of enrolled participants.

**Figure 4 ijerph-19-09975-f004:**
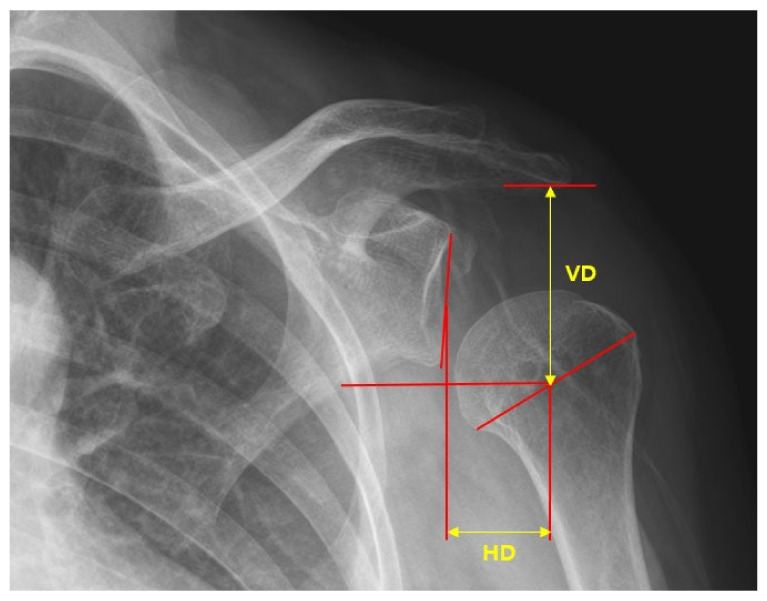
Measurement of horizontal distances (HD) and vertical distances (VD) in a true anteroposterior radiograph.

**Figure 5 ijerph-19-09975-f005:**
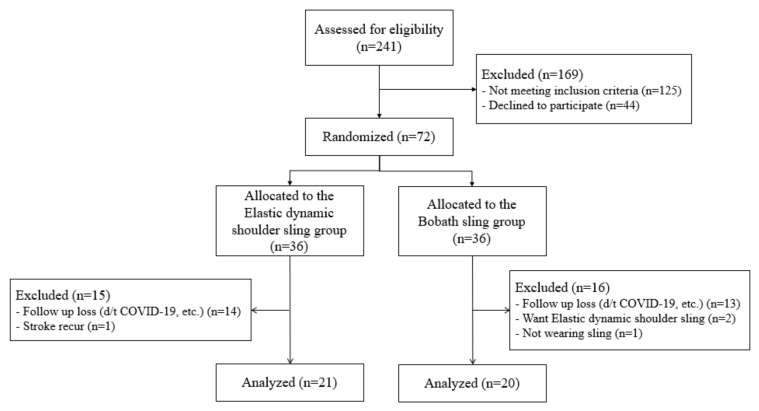
Flow chart of study participants enrollment.

**Table 1 ijerph-19-09975-t001:** Baseline characteristics of the elastic dynamic sling group and the Bobath sling group (mean ± standard deviations).

Characteristics	Elastic Dynamic Sling Group (*n* = 21)	Bobath Sling Group (*n* = 20)	*p* Value
Age (years)	64.76 ± 12.80	63.60 ± 14.46	0.774
Sex (male)	13	10	0.443
K-MBI	35.00 ± 17.85	30.90 ± 20.50	0.267
FMA-total	9.86 ± 9.94	8.10 ± 7.08	0.579
MMT (of shoulder)	1.44 ± 0.92	1.27 ± 0.76	0.413
Underlying disease			
HTN	81.0% (*n* = 17)	65.0% (*n* = 13)	0.247
DM	28.6% (*n* = 6)	30.0% (*n* = 6)	0.920
Dyslipidemia	14.3% (*n* = 3)	25.0% (*n* = 5)	0.387
Lesion			
Brain stem	4.76% (*n* = 1)	5% (*n* = 1)	0.972
Non-brain stem	95.24% (*n* = 20)	95% (*n* = 19)	
Stroke			
Infarction	71.43% (*n* = 15)	55% (*n* = 11)	0.275
Hemorrhage	28.57% (*n* = 6)	45% (*n* = 9)	

K-MBI, Korean-modified Barthel index; MMT, manual muscle testing; FMA, Fugl-Meyer assessment; HTN, hypertension; DM, diabetes mellitus.

**Table 2 ijerph-19-09975-t002:** Difference of vertical distance between the elastic dynamic sling group and the Bobath sling group.

	Average		*p*-Value
	Elastic Dynamic Sling Group (*n*= 21)	Bobath Sling Group (*n* = 20)	
VD_0_	42.99 ± 8.41	41.44 ± 9.06	
VD_1_	43.66 ± 8.26	44.57 ± 7.16	
VD_2_	45.60 ± 9.05	42.87 ± 9.16	
∆VD_1_	0.67 ± 7.76	3.05 ± 9.00	0.382
∆VD_2_	2.61 ± 10.95	1.43 ± 12.58	0.751

VD: vertical distance; D_0_: initial distance without sling; D_1_: 4 weeks after using sling; D_2_: 8 weeks after using sling; ∆VD_1_: difference between D_0_ and D_1_; ∆VD_2_: difference between D_0_ and D_2_; independent-samples *t*-test for between-group comparison.

**Table 3 ijerph-19-09975-t003:** Difference in horizontal distance between the elastic dynamic sling group and the Bobath sling group.

	Average		*p*-Value
	Elastic Dynamic Sling Group (*n* = 21)	Bobath Sling Group (*n* = 20)	
HD_0_	28.02 ± 2.66	27.44 ± 2.16	
HD_1_	27.13 ± 2.21	28.14 ± 2.52	
HD_2_	27.22 ± 2.40	29.73 ± 4.08	
∆HD_1_	−0.89 ± 2.46	0.48 ± 2.32	0.083
∆HD_2_	−0.80 ± 3.11	2.28 ± 3.66	**0.006**

Independent-samples *t*-test for between-group comparisons; HD: horizontal distance; D_0_: initial distance without sling; D_1_: 4 weeks after using sling; D_2_: 8 weeks after using sling; ∆HD_1_: difference between D_0_ and D_1_; ∆HD_2_: difference between D_0_ and D_2_.

**Table 4 ijerph-19-09975-t004:** Comparison of vertical and horizontal distances within the groups at four and eight weeks.

Measure	Baseline	4 Weeks	8 Weeks	*p*1-Value	*p*2-Value
	Mean ± SD	Mean ± SD	Mean ± SD		
Elastic Dynamic Sling Group					
Vertical Distance	42.99 ± 8.41	43.66 ± 8.26	45.60 ± 9.05	0.7355 ^a^	0.193 ^a^
Horizontal Distance	28.02 ± 2.66	27.13 ± 2.21	27.22 ± 2.40	0.584 ^a^	0.3309 ^a^
Bobath Group					
Vertical Distance	41.44 ± 9.06	44.57 ± 7.16	42.87 ± 9.16	0.1666 ^a^	0.4203 ^a^
Horizontal Distance	27.44 ± 2.16	28.14 ± 2.52	29.73 ± 4.08	**0.0273 ^a^**	**0.0023 ^a^**

Analysis was based on intention to treat. Values are presented as the average ± standard deviation. ^a^ Linear mixed model for within-group comparison; *p*1, comparison between baseline and four weeks; *p*2, comparison between baseline and 8 weeks.

**Table 5 ijerph-19-09975-t005:** Comparison of VAS, FMA-UE, MBI, MAS, and MMT within the groups at four and eight weeks.

Measure	Baseline	4 Weeks	8 Weeks	*p*1-Value	*p*2-Value
	Mean ± SD	Mean ± SD	Mean ± SD		
Elastic Dynamic Sling Group					
FMA-UE	7.52 ± 5.62	12.24 ± 7.45	15.25 ± 8.81	**0.0001 ^a^**	**<0.0001 ^a^**
FMA-Wrist	1.05 ± 1.94	2.05 ± 3.26	2.26 ± 4.06	0.0691 ^a^	**0.0045 ^a^**
FMA-Hand	1.14 ± 3.09	1.81 ± 2.89	2.85 ± 4.32	0.1081 ^a^	**0.0003 ^a^**
FMA-Co	0.14 ± 0.65	0.33 ± 1.06	1.45 ± 2.39	0.6379 ^a^	**0.0028 ^a^**
FMA-Total	10.05 ± 9.77	15.48 ± 13.28	22.15 ± 17.10	**0.0085 ^a^**	**<0.0001 ^a^**
MBI	35.00 ± 17.85	46.00 ± 17.98	58.80 ± 27.73	**0.0019 ^a^**	**<0.001 ^a^**
Pain (VAS)	1.52 ± 2.14	1.76 ± 2.47	1.86 ± 2.46	0.6332 ^a^	0.5046 ^a^
MAS	0.33 ± 0.58	0.67 ± 0.70	0.76 ± 0.87	**0.0444 ^a^**	**0.0109 ^a^**
MMT	1.55 ± 0.95	2.71 ± 1.88	3.19 ± 1.97	**<0.0001 ^a^**	**<0.0001 ^a^**
Bobath Group					
FMA-UE	6.70 ± 5.14	14.53 ± 8.52	16.65 ± 9.39	**0.0002 ^a^**	**<0.0001 ^a^**
FMA-Wrist	0.90 ± 2.10	2.26 ± 3.23	2.85 ± 3.38	**0.0071 ^a^**	**<0.0001 ^a^**
FMA-Hand	0.20 ± 0.62	2.58 ± 3.58	3.95 ± 4.87	**0.0109 ^a^**	**<0.0001 ^a^**
FMA-Co	0.30 ± 0.73	1.00 ± 1.73	1.00 ± 1.75	0.0522 ^a^	**0.0420 ^a^**
FMA-Total	8.10 ± 7.08	20.11 ± 14.91	24.60 ± 17.16	**0.0005 ^a^**	**<0.0001 ^a^**
MBI	30.90 ± 20.50	44.70 ± 22.75	51.30 ± 27.18	**0.0004 ^a^**	**<0.001 ^a^**
Pain (VAS)	1.35 ± 2.64	1.20 ± 1.82	1.70 ± 2.60	0.7717 ^a^	0.4994 ^a^
MAS	0.25 ± 0.53	0.48 ± 0.55	0.65 ± 0.90	0.1840 ^a^	**0.0211 ^a^**
MMT	1.35 ± 0.88	3.08 ± 1.66	3.35 ± 2.30	**<0.0001 ^a^**	**<0.0001 ^a^**

Analysis was based on intention to treat. Values were presented as the average ± standard deviation. ^a^ Linear mixed model for within-group comparison; *p*1, comparison between baseline and four weeks; *p*2, comparison between baseline and 8 weeks; FMA, Fugl-Meyer assessment scale; FMA-UE, upper extremity; FMA-Co, cooperation, MBI, modified Barthel index; MAS, modified Ashworth scale; VAS, visual analogue scale.

**Table 6 ijerph-19-09975-t006:** Comparison of VAS, FMA-UE, MBI, MAS, and MMT between groups at four and eight weeks.

	Average		*p*-Value
	Elastic Dynamic Sling Group (N = 21) Mean ± SD	Bobath Sling Group (N = 20) Mean ± SD	
4 weeks			
∆FMA-UE_1_	4.71 ± 4.66	7.68 ± 9.40	0.448
∆FMA-Wrist_1_	1.00 ± 2.17	1.32 ± 2.00	0.437
∆FMA-Hand_1_	0.67 ± 1.35	2.37 ± 3.48	0.063
∆FMA-Co_1_	0.19 ± 0.87	0.68 ± 1.63	0.146
∆FMA-Total_1_	5.43 ± 7.63	11.79 ± 15.06	0.125
∆MBI_1_	11.00 ± 12.32	13.80 ± 16.41	0.887
∆Pain (VAS)_1_	0.24 ± 2.21	−0.15 ± 1.79	0.908
∆MAS_1_	0.33 ± 0.86	0.23 ± 0.47	0.621
∆MMT_1_	1.17 ± 1.37	0.40 ± 0.94	0.246
8 weeks			
∆FMA-UE_2_	7.55 ± 6.71	9.95 ± 9.74	0.467
∆FMA-Wrist_2_	1.65 ± 3.25	1.95 ± 2.19	0.532
∆FMA-Hand_2_	1.65 ± 2.41	3.75 ± 4.84	0.329
∆FMA-Co_2_	1.30 ± 2.39	0.70 ± 1.53	0.585
∆FMA-Total_2_	11.80 ± 11.65	16.50 ± 16.20	0.377
∆MBI_2_	22.75 ± 17.27	20.40 ± 20.51	0.601
∆Pain (VAS)_2_	0.33 ± 2.61	0.35 ± 3.05	0.999
∆MAS_2_	0.43 ± 0.76	0.40 ± 0.94	0.839
∆MMT_2_	1.64 ± 1.41	2.00 ± 2.34	0.752

Analysis was based on intention to treat. Values were presented as the average ± standard deviation. Independent-samples *t*-test for between-group comparison; FMA, Fugl-Meyer assessment scale; FMA-UE, upper extremity; FMA-Co, cooperation, MBI, modified Barthel index; MAS, modified Ashworth scale; VAS, visual analogue scale.

## Data Availability

The data presented in this study are available on request from the corresponding author. The data are not publicly available due to privacy matters.

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
