# Peer review of "Elastic Dynamic Sling on Subluxation of Hemiplegic Shoulder in Patients with Subacute Stroke: A Multicenter Randomized Controlled Trial"

_ijerph, 2022, doi:10.3390/ijerph19169975_

Round 1
Reviewer 1 Report
Review: ijerph-1831549
Elastic Dynamic Sling on Subluxation of Hemiplegic Shoulder 2 in Patients with Subacute Stroke: A Multicenter Randomized 3 Controlled Trial
The article presents an interesting purpose to investigate whether there is a difference in the corrective effect, pain, and functional level of the hemiplegic upper extremity between patients with two different shoulder orthoses. This is very useful and original study. However, in my opinion, specific issues must be amended before acceptance.
Specific comments:
Comment #1: Abstract: I miss the highlighted sections Background, Purpose, Methods, Results and Conclusions
Lines 83-124
The structure of the sections is to much spread. In design section it’s only needed to explain only the design.
After the design section, in the “Participants” section must be explain the inclusion and exclusion criteria, as well the flowchart. In my opinion, the subsections of inclusion and exclusion criteria must be deleted.
Comment #2: Lines 130-145: The timeline of the assessing and follow-up could be explained by a figure or a table. Is needed to specify with pain scale was used. Moreover, it must be added a reference to support the use of each assessment method.
Comment #3: Lines 146-159
The figures of the slings must be referenced in the text.
Comment #4: Line 153-159: The authors state: “The disadvantages of the sling are that it laterally displaces the humerus into abduction, it is difficult to wear and remove it independently, and it lacks distal support; therefore, it can lead to increased hand edema or trauma [17-19]”. Therefore, careful monitoring of circulation…”. This sentence is more proper of a discussion section than the methods section.
Comment #5: Lines 134: Although the authors applied the manual strength assessment there is many other methods to assess patients’ strength with much more reliability and validity than the manual one. Moreover, authors should better explain which movements and muscles were tested with this method, as well as they explain the RX assessment
Comment #6: Line 217-218: In the results section the authors state: “The average age of the participants was 64.19 ± 13.48 years. The study population consisted of 26 patients with infarction, 15 with hemorrhage, two with brain stem lesions, and 39 with non-brain stem lesions”. This information is more proper for participants sections tan in the results because these are information that described the participants before the study start.
Comment #7: Section Results. In my opinion there are too many tables. Some results may be expressed better by figures than tables to enhance reading.
Comment #8: Discussion section must be improved. Discussion section should present the main findings of the study by the order that they appear in the results section. After that, the authors should add the reasons why they think these results are explained. The first and second paragraph of the discussion section are more proper of the introduction section.
Author Response
Thank you very much for taking time out of your busy schedule to review our paper. We are grateful for your comments. Please consider our response to each comment we received.
Please find below a copy of our point-by-point response to the comments. As you can see, we agreed with all the comments and have changed the manuscript accordingly. It was checked again by native speakers and minor corrections were made.
We take this opportunity to express our sincere gratitude to the reviewers who identified areas of our manuscript that needed modification and improvement. We hope that we have adequately responded to all of the comments and that the revised manuscript is acceptable for publication in IJERPH.

Reviewer 2 Report
Dear Authors,
Thank you for the opportunity to review your manuscript. I recommend structuring the abstract. I recommend following the CONSORT guidelines.
L19 “corrective effect is not an appropriate term” and then in the goal the subluxation disappears, is it only the hemiplegic shoulder pain?: The aim of this study was to investigate the effect of two different orthoses on pain and functioning in hemiplegic shoulder subluxation.
L21 In the methods better characterize the eligibility of the population not the number of patients enrolled (which is a selection result). Describe the evaluation moment but the timepoint is missing.
L26 In the results you have to include some objective data, above all the differences in pain and function (how large? how significant?), Among other things, the objective does not speak of imaging as an outcome but of pain and functioning, in the methods we do not speak of distance but of "true" x-ray (among other things, what is meant by true?)
I do not recommend using the term corrective effect. it is improper and there is no scale on the subject that provides a correction index, we are talking about pain and functioning, does one of the two interventions improve pain or functioning? also because in conclusion even if there was an improvement in imaging but there would be no significant clinical difference, the conclusions are clear.
Introduction
L41 But not always.
L43 I would suggest inserting: "The etiopathogenesis is unclear, but it has been suggested that"
L51 Moreover, shoulder subluxation might enhance pain and joint contracture impacting negatively on the recovery of upper-extremity function in patients with stroke
I must state that: Stroke can lead to hemiplegic shoulder pain or not. Hemiplegic shoulder pain may underlie a shoulder subluxation or not. There may also be no pain or functional deficit in a hemiplegic shoulder, as all phenomena could coexist. All this is not clear in the introduction.
L61 Botulinum toxin, PENS, TENS, NMES https://doi.org/10.1016/j.rehab.2021.101602
Methods
L114-128 RESULTS.
L125: I recommend following the CONSORT guidelines. The interventions for each group with sufficient details to allow replication, including how and when they were actually administered.
Completely defined pre-specified primary and secondary outcome measures, including how and when they were assessed
L179 Data analysis is always part of an outcome, among other things it is not clear what the primary outcome is
L205 A sample size calculation is missing, an effect size calculation is missing
RESULTS
Put the consort flowchart first, explaining the whole enrollment process, then the complete characteristics of the patients included. in this regard but… the pain felt was similar before the intervention?
Table 4 EDSS group?
DISCUSSION
L267 By convention you need to paraphrase the study objective before repeating the major study findings.
Shoulder subluxation is a “possible” complication. Anyway, In discussion… discuss the results of your study. Discuss how literature might legitimize or subvert their findings. Why did the two interventions have similar results? Some authors do not justify any use of orthoses .. are there perspectives, glimmers ?, perhaps as oriented to a temporary treatment?
L267-280 Much of this has already been pointed out in the introduction, just provide it to discuss the study results
L286- Much better, but always in light of the study results
L331 Why and what justification do you give?
L336 This is why x-ray measures cannot be a primary outcome
Author Response
Thank you very much for taking time out of your busy schedule to review our paper. We are grateful for your comments. Please consider our response to each comment we received.
Please find below a copy of our point-by-point response to the comments raised by the reviewers. As you can see, we agreed with all the comments and have changed the manuscript accordingly. It was checked again by native speakers and minor corrections were made. The modified text appears underlined in the revised manuscript.
We take this opportunity to express our sincere gratitude to you and the reviewers who identified areas of our manuscript that needed modification and improvement. We hope that we have adequately responded to all of the comments and that the revised manuscript is acceptable for publication in IJERPH.

Round 2
Reviewer 1 Report
I appreciate the good review of the draft according the suggestions. The article has improved very much.
Reviewer 2 Report
Dear Authors, I can suggest your manuscript's suitability for publication